# Antitumoral Activity of *Leptocarpha rivularis* Flower Extracts against Gastric Cancer Cells

**DOI:** 10.3390/ijms24021439

**Published:** 2023-01-11

**Authors:** Nicolás Carrasco, Maritza Garrido, Iván Montenegro, Alejandro Madrid, Ricardo Hartley, Iván González, Mariaignacia Rubilar, Joan Villena, Manuel Valenzuela-Valderrama

**Affiliations:** 1Laboratorio de Microbiología Celular, Instituto de Investigación y Postgrado, Universidad Central de Chile, Lord Cochrane 418, Santiago 8330546, Chile; 2Laboratorio de Endocrinología y Biología de la Reproducción, Hospital Clínico Universidad de Chile, Santiago 8380456, Chile; 3Centro de Investigaciones Biomédicas, Escuela de Obstetricia, Universidad de Valparaíso, Angamos 655, Reñaca, Viña del Mar 2340000, Chile; 4Laboratorio de Productos Naturales y Síntesis Orgánica (LPNSO), Departamento de Química, Facultad de Ciencias Naturales y Exactas, Universidad de Playa Ancha, Avda. Leopoldo Carvallo 270, Playa Ancha, Valparaíso 2340000, Chile; 5Departamento de Química, Facultad de Ciencias Naturales, Matemática y del Medio Ambiente, Universidad Tecnológica Metropolitana, Las Palmeras 3360, Ñuñoa, Santiago 7800003, Chile; 6Centro de Investigaciones Biomédicas, Escuela de Medicina, Universidad de Valparaíso, Angamos 655, Reñaca, Viña del Mar 2340000, Chile

**Keywords:** gastric cancer, *Leptocarpha rivularis* flowers, leptocarpin, organic extracts, invasion, migration, clonogenicity, cytotoxicity, natural compounds

## Abstract

*Leptocarpha rivularis* is a native South American plant used ancestrally by Mapuche people to treat gastrointestinal ailments. *L. rivularis* flower extracts are rich in molecules with therapeutic potential, including the sesquiterpene lactone leptocarpin, which displays cytotoxic effects against various cancer types in vitro. However, the combination of active molecules in these extracts could offer a hitherto unexplored potential for targeting cancer. In this study, we investigated the effect of *L. rivularis* flower extracts on the proliferation, survival, and spread parameters of gastric cancer cells in vitro. Gastric cancer (AGS and MKN-45) and normal immortalized (GES-1) cell lines were treated with different concentrations of *L. rivularis* flower extracts (DCM, Hex, EtOAc, and EtOH) and we determined the changes in proliferation (MTS assay, cell cycle analysis), cell viability/cytotoxicity (trypan blue exclusion assay, DEVDase activity, mitochondrial membrane potential MMP, and clonogenic ability), senescence (β-galactosidase activity) and spread potential (invasion and migration assays using the Boyden chamber approach) in all these cells. The results showed that the DCM, EtOAc, and Hex extracts display a selective antitumoral effect in gastric cancer cells by affecting all the cancer parameters tested. These findings reveal an attractive antitumoral potential of *L. rivularis* flower extracts by targeting several acquired capabilities of cancer cells.

## 1. Introduction

Cancer continues to be one of the biggest health challenges around the world. In fact, the death toll due to neoplasia reached almost 10 million in 2020 [1]. Although significant advances have been made in molecular diagnosis and personalized medicine approaches, therapeutic options remain limited compared to the diversity of genetic alterations, the appearance of resistance, and the degree of tumor aggressiveness, even among those that diverge from the same primary tumor [2,3,4]. In this regard, gastric cancer does not represent an exception in this panorama; on the contrary, it shows high mortality, generally diagnosed in late stages when there is evidence of metastasis and therapeutic options are limited [5,6,7]. Currently, resection of the affected region of the stomach is accompanied by a preoperative or postoperative adjuvant chemotherapy based on the combined administration of fluoropyrimidines, platinum compounds, taxanes, and monoclonal antibodies (Trastuzumab), considering the HER2 status of the patients [5,7]. Although first-line therapy achieves tumor remission if detected early [8], disease-free survival is reduced in the case of advanced tumors, which poorly respond to conventional treatments due to the emergence of metastasis and drug resistance [9,10].

First-line therapy drugs such as 5-fluorouracil or cisplatin exert their action by inhibiting nucleic acid metabolism [11,12]. The taxane paclitaxel blocks cells in the G2/M phase of the cell cycle by preventing the assembly of tubulin into microtubules [13], and Trastuzumab blocks HER2-mediated proliferative signaling [14]. As noted, these alternatives are focused on causing DNA replication or mitosis arrest, leading to a cell collapse that ultimately results in apoptosis [11,12]. In this scenario, current therapy fails to attack quiescent malignant cells, such as gastric cancer stem cells [10,15] or those that have acquired chemoresistance [16], and even those processes implicated in the spread of malignant cells [17].

The use of natural compounds and plant extracts has emerged as a plausible strategy against cancer [18,19,20], particularly gastric cancer [21,22,23,24]. For instance, the alkaloid vincristine derived from plants of the genus *Vinca* (Oncovin^®^) [25] and the taxane paclitaxel (taxol^®^) derived from *Taxus baccata*, are both used against gastric neoplasms [24]. Bioactive compounds found in food, including alkaloids, flavonoids, phytosterols, terpenoids, and polyphenols, also have a recognized potential for preventing and treating gastric cancer [26].

*Leptocarpha rivularis* (“Palo negro”) is a plant native to South America, which grows mainly in the southern part of Chile [27]. It has been used ancestrally by people of the Mapuche ethnic group to treat gastrointestinal ailments [28]. *L. rivularis* produces a variety of bioactive compounds, including a group of molecules termed sesquiterpene lactones (SLs) [29]. SLs have attracted the attention of researchers for exhibiting anticancer, antifungal, anti-inflammatory, antimicrobial, and antioxidant activities [30]. One of the most abundant SLs in *L. rivularis* is leptocarpin [31]. This molecule displays several properties, including antitumoral effects on different cancer cell lines, with low toxicity and high specificity [32]. Moreover, a previous study described the composition of *Leptocarpha rivularis* flower extracts [29], showing the presence of triterpenes, triterpenoids, and other compounds with reportedly antiproliferative, pro-apoptotic, anti-invasion, and anti-angiogenic activity, such as dehydrocostus lactone, thunbergol, caryophyllene oxide, intermedeol, lupeol, beta-amyrin, leptocarpin and thujone, among others [30]. Of note, leptocarpin is only present in the EtOAc extract [29].

Here, we investigated the antitumoral effects of the *L. rivularis* flower extracts: dichloromethane (DCM), ethyl acetate (EtOAc), n-hexane (Hex), and ethanol (EtOH) against gastric cancer cells in vitro by evaluating their effect on some cancer-related parameters involved in replicative immortality, resistance to cell death, angiogenesis and spread (migration and invasion). Our results showed that the DCM, EtOAc, and Hex *L. rivularis* flower extracts are selective against gastric cancer cells (AGS and MKN-45) and attenuate their malignant potential. Interestingly, these effects were comparable to those produced by purified leptocarpin, even in extracts where this compound was not present (DCM and Hex). Together, these data expand our knowledge about the antitumoral potential of the combination of natural compounds found in *L. rivularis* flower extracts, particularly against gastric cancer.

## 2. Results

### 2.1. Extract Yields

The sequential Soxhlet extraction method was used to obtain flower extracts using Hex, DCM, EtOAc, and EtOH solvents. The yields achieved were 6.50% (*w*/*w*), 5.80% (*w*/*w*), 4.60% (*w*/*w*), and 15.52% (*w*/*w*), respectively.

### 2.2. Phytochemical Content of L. rivularis Flower Extracts

*L. rivularis* flower extracts, Hex, DCM, and EtOAc were analyzed by chromatographic analysis (GC/MS). The results obtained for this analysis are shown in Table 1, Table 2 and Table 3. The Hex extract contained principally fatty acids and derivatives (see Table 1). The DCM extract had thunbergol, intermedeol and the triterpenoids ß-Amyrin acetate and lupeol acetate (see Table 2). The EtOAc extract contained sesquiterpene lactones, including Leptocarpin and alcohol duvatrienediol, the sterol stigmasterol, and fenretinide compounds, among others (see Table 3).

### 2.3. Antiproliferative Effect of L. rivularis Flower Extracts on Gastric Cell Lines

In this study, we wondered if the natural compounds present in these extracts might offer a still unrevealed potential against gastric pathologies, particularly cancer. Thus, we were interested in investigating the antitumoral effect of such extracts on gastric cancer-derived cell lines by evaluating several relevant parameters involved in cancer development and spread, such as clonogenic ability, angiogenesis invasion, migration, and senescence. Initially, we explored the effect of some *L. rivularis* flower extracts (DCM, EtOAc, Hex, and EtOH) on the proliferation of the human gastric epithelial cancer cell lines AGS and MKN-45. We also included the normal immortalized human gastric epithelial cell line GES-1 as a control. As shown in Figure 1A, the proliferation of the three cell lines was dose-dependently inhibited by all extracts tested after 24 h or 48 h of treatment. A similar effect was displayed by leptocarpin, a sesquiterpene lactone abundant in the EtOAc extract, which reportedly shows antitumoral activity [18,19]. To compare these effects quantitatively, we calculated each extract’s half-maximal inhibitory concentration (IC_50_). As shown in Figure 1B, the extracts displayed comparable IC_50_ values; however, the most polar extract, EtOH, was considerably less active than the others. On the other hand, leptocarpin was slightly more bioactive than the extracts, producing lower IC_50_ values in these assays (IC_50_ values between 5.10 ± 3.88 µg/mL, up to 8.10 ± 0.38 µg/mL). Additionally, we calculated the selectivity index of each extract in AGS and MKN45 cells. Taking into account that a selectivity index (SI) > 2 is considered a good value [34], our results indicated that in terms of proliferation, these extracts did not display a more antiproliferative effect in the cancer cell lines compared with the normal cell line (Table 4). Despite this conclusion, the EtOAc extract exhibited higher SI values against AGS and MKN45 cells.

Furthermore, using the trypan blue exclusion assay, we evaluated the viability of these gastric cell lines following treatment with the extracts or leptocarpin as a control. Although it is known that this assay does not differentiate between different forms of cell death, it can detect the loss of plasma membrane integrity, which is a point of no return during the cell death process [23]. As shown in Figure 1C, the treatments did not dramatically increase trypan blue permeability; however, a significant drop in cell viability (up to 30%) was observed at 48 h following treatment with DCM or EtOAc extracts. For comparison purposes, the cells were also treated with the chemotherapeutic agent Etoposide (50 µM), which produced a robust cytotoxic effect in the three cell lines. Next, we evaluated cell membrane permeability using a more sensitive indicator of cytotoxicity: the combination between Ethidium homodimer-1 (EthD-1) and Calcein-AM. EthD-1 enters cells with damaged plasma membranes and undergoes a fifty-fold enhancement of red fluorescence upon binding to nucleic acids. Calcein-AM is retained and hydrolyzed by esterases in live cells, producing an intense green fluorescence. As shown in the panels of Figure 1D, red fluorescence was evident in AGS cells treated with every extract; however, this was not evident for GES-1 cells treated with the extracts.

Finally, we complemented these observations by determining changes in the cell cycle distribution of GES-1 and AGS cells in response to DCM and EtOAc extracts, two of the most bioactive extracts in these cells. As shown in Figure 1E, the antiproliferative effect of *L rivularis* flower extracts was also reflected in changes in cell cycle distribution, characterized by the accumulation of cells arrested in the S or G2/M phases. Together, these results reveal the antiproliferative effect of the *L. rivularis* flower extracts on normal immortalized and cancer gastric cell lines but a differential cytotoxic effect in gastric cancer cells, evidenced by cell membrane permeabilization to EthD-1.

### 2.4. L. rivularis Flower Extracts Affect Mitochondrial Membrane Potential and Induce Caspase Activity in Gastric Cancer Cells

Our results showed that the *L. rivularis* flower extracts inhibited the proliferation of normal and gastric cancer cells, exerting a cytotoxic effect on cancer cells. Thus, we wanted to gain insight into the perturbed stress pathways responsible for these observations. As a first approach, we treated GES-1 and AGS cells for 24 h with 20 µg/mL of organic extracts or FCCP 1µM, a mitochondrial oxidative phosphorylation uncoupler used as a positive control. Then, the loss of mitochondrial membrane potential was assessed following Rhodamine 231 fluorescence by flow cytometry [35]. As shown in Figure 2A, histograms revealed a robust loss of the Rhodamine-positive population in AGS cells, where the treatment with DCM, EtOAc, and Hex extracts depolarized more than 95% of the cells. Notably, GES-1 cells retained a considerable percentage of the Rhodamine-positive population (>44%) under similar treatments (Figure 2B). In agreement with our previous results, the EtOH extract did not significantly affect the mitochondrial membrane potential in both cell lines (Figure 2B), and FCCP 1µM produced a similar differential effect. The effect of a lower concentration of extracts (2.5 µg/mL) is also shown for comparison.

Mitochondria integrate apoptotic signals from both the intrinsic and extrinsic pathways, where membrane potential alterations are key determinants in executing apoptosis [36]. Since depletion of mitochondrial membrane potential causes the release of pro-apoptotic factors, such as cytochrome c, triggering caspase activation [37], we investigated the effect of extracts on caspase activity. Thus, we determined DEVDase enzymatic activity (caspase 3/7 activity) in gastric cells, which is activated by many apoptotic signals [37] and primary executors of apoptosis by inducing oligonucleosomal DNA fragmentation, among other events [38]. As shown in Figure 2C, DEVDase activity significantly increased in AGS cells following treatment with DCM, EtOAc, or Hex extracts (10 µg/mL) but not with the EtOH extract. Interestingly, DEVDase activity was not significantly affected in GES-1 cells, despite the positive control Etop 50 µM.

Next, by Western blot, we analyzed changes in the protein levels of the cleaved pro-apoptotic enzymes Caspase-3 and Caspase-9, the DNA repair protein PARP-1, the transcriptional regulator p53, the antiapoptotic protein Survivin, and the antioxidant/anti-inflammatory enzyme HO-1. As shown in Figure 2D, a qualitative analysis of the blots showed that cleaved caspase-3 was only evident in AGS cells treated with the Hex extract. On the other hand, cleaved caspase-9 was not detectable in response to any of the extracts, except when Etoposide was used. These results agree with the mild induction of DEVDase activity in response to the extracts, compared with the outcome produced by Etoposide. On the other hand, the extracts did not cause an appreciable change in the protein levels of cleaved PARP-1, p53, or Survivin.

Interestingly, HO-1 was consistently induced in both cell lines in response to all extracts tested, but slightly by Etoposide. This enzyme exerts a cytoprotective role against oxidative injury and other cellular stresses [39]. Together, these results indicate that *L. rivularis* flower extracts produce a cytotoxic effect characterized by loss of mitochondrial membrane potential, activation of Caspase-3, and induction of HO-1. These changes are consistent with oxidative stress-induced cytotoxicity, which could affect normal and cancer cells. However, our results show that a discriminating effect is achieved by modulating the concentration of the extracts.

### 2.5. Effect of L. rivularis Flower Extracts on the Clonogenic Ability of Gastric Cells

Considering that *L. rivularis* flower extracts inhibited proliferation and induced apoptosis in gastric cancer cells, we wondered if this cytotoxic effect could be accompanied by a decrease in the malignant potential of these cells. Thus, we performed an in vitro cell survival approach known as clonogenic assay or colony formation assay, which can assess the effect of a cytotoxic agent on the ability of a single cell to proliferate indefinitely and grow into a colony. With this in mind, we treated AGS and MKN-45 cells with *L. rivularis* flower extracts for 24 h and performed the assay. Additionally, leptocarpin and Etoposide (Etop) were included as controls. As shown in Figure 3A,B, AGS and MKN-45 cells significantly reduced their clonogenic potential in response to treatment with the extracts, leptocarpin, or Etoposide, except when treated with the EtOH extract, which did exert a significant effect. Of note, this inhibitory effect was more pronounced in AGS than in MKN-45 cells. For instance, DCM extract and leptocarpin treatments inhibited clonogenicity in AGS cells by nearly 70% and 85%, respectively, vs. 40% and 50% in MKN-45 under similar conditions. On the other hand, the immortalized gastric cell line GES-1 was affected only by leptocarpin, DCM, and EtOAc; however, such inhibition was notably less pronounced (30%, 35%, and 10%, respectively) than that observed in the gastric cancer cell lines. As expected, Etoposide displayed a strong inhibitory effect in all cell lines tested.

These observations are consistent with our previous results showing a differential cytotoxic effect between the gastric cancer cells and the normal immortalized cell line GES-1.

### 2.6. L. rivularis Flower Extracts Induced a Senescent Phenotype in Gastric Cancer Cells

Senescence is a state of permanent cell cycle arrest in response to damaging stimuli, such as DNA damage, oncogene activation, oxidative stress induction, chemotherapy, lysosome or endoplasmic reticulum stress, mitochondrial dysfunction, and dysregulation of epigenetic marks, among others [40]. Physiologically, it is crucial to promote tissue remodeling during embryonic development and after injury [41]. Even though a specific senescence marker has not been described, an increase in lysosomal capacity is characteristic of senescent cells [42]. In turn, this is linked to a rise in the activity of the lysosomal enzyme, senescence-associated beta-galactosidase (SA-βgal), which is commonly used as a marker of the accumulation of lysosomal content in senescent cells [42]. Since extracts from *L. rivularis* flowers considerably decreased the clonogenic capacity of gastric cells (Figure 3), we wondered if this effect was also accompanied by the induction of senescence in these cells. For this purpose, we treated AGS and MKN-45 cells with sublethal concentrations of the extracts for 72 h. As expected, SA-βgal activity was evidenced by the accumulation of the green fluorescent product (white arrows in Figure 4). Although qualitative, the green signal was more evident in cells treated with EtOAc and Hex extracts. This result confirms that *L. rivularis* flower extracts target the unlimited proliferation capacity of gastric cancer cells. 

### 2.7. L. rivularis Flower Extracts Reduced Migration and Invasion Potential in Gastric Cancer Cells

Next, we wanted to evaluate the effect of the extracts on other parameters related to tumor aggressiveness, such as the ability of tumor cells to migrate from the primary tumor and invade other distant tissues. To this end, we performed migration and invasion assays using Boyden’s chamber approach [43,44]. As seen in Figure 5A, treatment with the *L. rivularis* flower extracts significantly reduced the migration of AGS and MKN-45 cells, except for the EtOH extract. Similarly, the extracts also reduced the invasion potential of both cell lines (Figure 5C); however, for MKN-45 cells, the most significant reduction was achieved with the EtOAc extract, which reduced their invasion potential by only about 30%. The EtOAc and DCM extracts also reduced the migration and invasion of GES-1 cells, but the Hex and EtOH extracts did not significantly affect these parameters. For visual evaluation, representative photographs of the crystal violet-stained cells that migrated and invaded the underside of the Transwell^®^ inserts in response to the different conditions are shown in Figure 5B,D, respectively. Of note, the percentage of inhibition achieved by treatment with the extracts was comparable to that obtained with purified leptocarpin. Together, these results show that the organic extracts effectively reduce the migration and invasion potential of gastric cancer cells. Furthermore, Hex and DCM extracts were able to discriminate between cancer and normal cells in relation to migration and invasion, respectively. As expected, Etop 50 µM produced a robust inhibitory effect in all cell lines tested.

### 2.8. The EtOAc Extract of L. rivularis Flowers Reduced the Vasculogenic Capacity of EA.hy926 Cells

The generation of new blood vessels is crucial for the tumor to acquire the nutrients and oxygen necessary to sustain its growth [45]. Thus, the induction of angiogenesis is a hallmark of cancer [46]. For this reason, we wondered whether the extracts from *L. rivularis* flowers could affect the angiogenic capacity of an in vitro cellular model of vascularization. For this purpose, we treated EA.hy926 human endothelial cells with one of the most active extracts (EtOAc) and leptocarpin, to assess changes in the angiogenic index of these cells using the Matrigel Vasculogenic Assay, which is based on the quantification of interactions that are characteristic of an angiogenic process, such as the formation of polygonal structures (meshes) and junctions [47]. First, we evaluated the cytotoxicity produced by the EtOAc extract and leptocarpin using the MTS assay. As shown in Figure 6A, the EtOAc extract inhibited the viability of EA.hy926 cells in a dose-dependent manner (IC_50_ of 1.79 ± 0.39 µg/mL and 1.26 ± 0.042 µg/mL at 24 h or 48 h, respectively). On the other hand, the cytotoxicity produced by leptocarpin was less pronounced (IC_50_ 4.75 ± 0.33 µg/mL and 3.9 ± 0.34 µg/mL at 24 and 48 h, respectively).

Next, we evaluated the effect of the EtOAc extract or leptocarpin on the angiogenic index of EA.hy926 cells using low concentrations of both agents to avoid the impact related to their cytotoxicity. As shown in Figure 6B, the quantification of the morphological parameters (see Figure 6C) of the cells treated with the EtOAc extract or leptocarpin at concentrations of 1.0 µg/mL and 2.5 µg/mL showed that both conditions were able to decrease the angiogenic index of EA.hy926 cells significantly. Cells were cultured in the presence of VEGF 20 ng/mL, which also induced angiogenesis, as expected.

## 3. Discussion

Flowers contain numerous phytochemical constituents, many of which are responsible for diverse biological activities, including cancer development. In this sense, this study contributes to this field by providing data concerning the cytotoxic effect of *L. rivularis* flower extracts against gastric cancer cell lines.

Cancer, one of the leading causes of death in the world [48], arises from a stepwise accumulation of genetic or epigenetic changes that liberate neoplastic cells from the homeostatic regulation that controls normal cell proliferation [49]. Chemotherapy is the most common and effective treatment, but its side effects are often severe [50]. Research and development of new antitumoral compounds are needed.

The results obtained in this study reveal the antitumoral effect of *L. rivularis* flower extracts on gastric cancer cells. Importantly, our findings show that the extracts decreased the malignant potential of gastric cancer cells by targeting several vital features required for cancer progression. Notably, the extracts caused the arrest of the cell cycle, loss of mitochondrial membrane potential and Caspase-3 activation, attenuation of clonogenic potential, and induction of a senescent phenotype (Figure 1, Figure 2, Figure 3 and Figure 4, respectively). In addition, the extracts decreased the migration and invasion potential in AGS and MKN-45 cells, both critical features for the metastatic spreading of malignant cells. On the other hand, and although we did not demonstrate it directly in gastric cancer cells, the anti-angiogenic effect of the extracts in the EA.hy926 endothelial cell line (Figure 6) suggests that these extracts could also be effective in attacking the neovascularization process in gastric cancer.

An important observation to highlight is that the extracts produced a significant antiproliferative effect against normal and cancer-derived gastric cells. In this sense, the IC_50_ was quite similar between the different cell lines tested, which was also reflected in low SI values (Table 3). This unexpected result seems to be particular for gastric cells since these extracts showed SI values > 2 in other cell types, such as breast (MCF-7), prostate (PC-3) and colon (HT29) [29]. Moreover, similar findings were obtained using purified leptocarpin (Table 3). Despite these results, we used concentrations close to the IC_50_ for each extract (DCM 5 ug/mL, EtOAc 5 ug/mL, Hex 10 ug/mL, and EtOH 15 ug/mL) to evaluate several parameters related to the malignant potential of gastric cancer cells and also ensure consistent biological effects. In this sense, the activity of the extracts we used was able to discriminate between cancer (AGS and MKN-45) and normal immortalized GES-1 gastric cells, at least in cytotoxicity. For example, loss of mitochondrial membrane depolarization and induction of Caspase-3 activity was significantly less evident in GES-1 cells (Figure 2A,C, respectively). Of note, the proliferation assays, such as MTS, do not discriminate whether the observed effect is a consequence of inhibition of proliferation itself or is a phenomenon that includes cell death, limiting the use of SI values to assess the antitumoral impact of specific compounds. In agreement with this notion, our results show that the extracts produced a dose-dependent antiproliferative effect in GES-1 cells; however, apoptosis was not evident in response to the concentration of extracts indicated. This observation contrasts with the effect produced by Etoposide, a classical chemotherapeutic agent, which had a robust cytotoxic effect in all cell lines tested (Figure 1C,D, Figure 2C, and Figure 3). Therefore, these extracts may offer an interesting potential for gastric cancer treatment, thus avoiding the adverse effects of conventional antineoplastic molecules.

Members of the family of *Asteraceae* are rich in compounds such as polyacetylenes, diterpenes and sesquiterpene lactones (SQL), which display various biological activities, including antitumoral, anti-inflammatory, neurocytotoxic, and cardiotonic effects [51]. In this study, we found by GC–MS analysis that *L. rivularis* flower extracts are rich in fatty acids derivatives, sesquiterpenes, sesquiterpene lactones, triterpenes, triterpenoids. Among them, we found compounds with antineoplastic activity, such as duvatrienediol, dehydrocostus lactone, thunbergol, lupeyl acetate, beta-amyrin acetate, leptocarpin, and thujone, stigmasterol and fenretinide [29,52,53,54,55,56]. Of note, the antitumoral activity of the extracts was generally comparable to that observed for purified leptocarpin. This observation is remarkable, since a possible alternative therapy based on these extracts could dispense with costly purifications of active compounds. Furthermore, a combination of several relevant compounds could compensate for the inefficacy of a single molecule against a particular parameter; for instance, the EtOAc extract (1.0 µg/mL and 2.5 µg/mL) was able to inhibit the vasculogenic index of EA.hy926 cells significantly; however, leptocarpin at the same concentrations did not. With this respect, EtOAC contains a significant concentration of stigmasterol, a sterol inhibits angiogenesis in human umbilical vein endothelial cells [55].

Leptocarpin exhibits significant cytotoxic activity against cancer cell lines and selectivity index (SI) values between 2.9 and 4.9 [32]. However, and although it is abundant in flowers, leptocarpin is only extracted by EtOAc [29], which indicates that the antitumoral effect of the extracts is not limited to the presence of leptocarpin, but to the existence and nature of contributing molecules with different biological activities. Thus, the antitumoral activity of *L. rivularis* flower extracts in gastric cancer cells is likely a synergistic effect caused by the most abundant compounds of these extracts. Moreover, other studies have demonstrated that some of these compounds (dehydrocostus lactone and thujone) induce apoptosis [57,58]. In addition, dehydrocostus lactone possesses various biological activities, including anti-inflammatory [59], antioxidant [60], and anticancer activities [61]. Research has associated the anticancer activities of dehydrocostus lactone to the inhibition of cancer cell proliferation and induction of cancer cell apoptosis [53], inhibition of migration and invasion [62], and inhibition of angiogenesis [63]. On the other hand, α-thujone and beta-thujone have been used to treat many diseases due to antioxidant activity [58,64,65]. Fenretinide is a retinoic acid analog that exhibits several antitumoral activities in preclinical and clinical studies on several cancer types [66,67]. Finally, another abundant compound of these extracts, the triterpenes α and β amyrin acetate, are recognized cytotoxic compounds [68].

Other cellular processes related to tumor aggressiveness are the migration and invasion capacities that allow tumor cells to reach distant tissues [69]. Effects on migration and invasion have been described for some of the compounds mentioned previously in these extracts, for instance, dehydrocostus lactone and lupeyl acetate [62,70]. In the case of angiogenesis, a key process that enables tumor growth [58], thujone and lupeyl acetate inhibited this process in other cancer cell lines [55,58]. Of course, cell spreading and neovascularization are physiological processes in adult tissues required for tissue repair [71,72]. This point is relevant, given that migration and invasion parameters were also affected to some extent in normal GES-1 gastric cells (Figure 5). Therefore, we cannot exclude the possibility that these extracts could eventually alter relevant physiological functions. This notion needs to be addressed using in vivo models to evaluate possible hormetic effects and future experiments should address this relevant issue.

The findings of our study provide evidence that DCM, EtOAc, and Hex extracts of *L. rivularis* flowers display an antitumoral effect on gastric cancer-derived cell lines by targeting relevant processes required for tumor growth and spread, such as clonogenicity, senescence, invasion, migration, and angiogenesis. Such effects can be ascribed to a combination of bioactive compounds present in these extracts; however, further investigation is necessary to establish the role of a particular molecule and determine if this effect occurs in a synergic manner.

## 4. Materials and Methods

### 4.1. Obtention of Leptocarpha rivularis Flower Extracts

The compounds were exhaustively extracted from dried powdered flowers of *L. rivularis* (200 g) with 70% ethanol (300 mL), using a Soxhlet extractor for 16 h at 50 °C. The extract was evaporated and then differentially extracted with ethyl acetate (EtOAc), dichloromethane (DCM), n-hexane (Hex), or ethanol (EtOH), using an orbital shaker (170 rpm) at 25 °C for 72 h, as previously described [29], using chromatographic grade solvents (Sigma-Aldrich, Darmstadt, Germany). The resulting extract was filtered through Whatman No. 1 filter paper (Sigma-Aldrich) and dried under reduced pressure with a rotatory evaporator (Rotavapor R-300, BÜCHI, Barcelona, Spain). Finally, each extract was weighed, and the yield was calculated. *L. rivularis* flower extracts were kept at −4°C before further analyses. Leptocarpin (molecular weight 362 g/mol) was purified from *L. rivularis* as previously described [32]. Work solutions of each extract were prepared in 50% ethanol at a final concentration of 10 mg/mL and stored at −80 °C until used. For all experiments performed, the control condition included 0.1% ethanol.

#### Chromatographic Analysis

The Hex, DCM, and EtOAc extracts were diluted with chloroform and analyzed by gas chromatography (Hewlett Packard, Palo Alto, CA, USA) as previously described [73]. The operating conditions were as follows: on-column injection; injector temperature, 250 °C; detector temperature, 280 °C; carrier gas, He at 1.0 mL/min; oven temperature program: 40°C increase to 260 °C at 4°C/min, and then 260 °C for 5 min, to afford the best separation through a capillary Rtx-5MS column. The mass detector ionization employed an electron impact of 70 eV. Compounds in the chromatograms were identified by comparing their mass spectra with those found in the NIST/EPA/NIH mass spectral Library [74]. Chromatographic peaks were considered “unknown” and discarded in this identification process when their similarity index (MATCH) and reverse similarity index (RMATCH) were less than 850 [75]. These parameters refer to the degree the target spectrum matches the standard spectrum in the NIST Library (the value 1000 indicates a perfect fit), and by comparison of their retention index with those reported in the literature [33], for the same type of column or those of commercial standards, when available. The retention indices were determined under the same operating conditions compared to a homologous n-alkanes series (C8–C36) by the equation: RI = 100 × (n + Tr(unknown) − Tr(n)/Tr(N) − Tr(n))(1) where, n = the number of carbon atoms in the smaller n-alkane; N = the number of carbon atoms in the larger n-alkane; and Tr = the retention time. Relative concentration of compounds was calculated using the peak area normalization

### 4.2. Cell lines and Culture Conditions

The human gastric cancer-derived cell lines AGS (CRL-1739) and MKN-45 (JCRB0254) were obtained from the American Type Culture Collection (ATCC, Manassas, VA, USA) and JCRB Cell Bank (Osaka, Japan), respectively. The human immortalized gastric cell line GES-1 was kindly gifted by Dr. Dawit Kidane (The University of Texas at Austin, USA). The immortalized human umbilical vein cell line EA.hy926 (CRL-2922) was obtained from ATCC. AGS and MKN-45 cells were cultured in RPMI 1640 medium, GES-1 cells were cultured in DMEM high glucose medium (ThermoFisher Scientific, Waltham MA, USA), and EA.hy926 cells were cultured in DMEM Ham F12 without red phenol (Thermo Fisher Scientific). All of them were supplemented with 2 mM glutamine, 10% fetal bovine serum (Biological Industries, Beit-Hamek, Israel), and antibiotics (100 U/mL penicillin and 100 µg/mL streptomycin) in a humidified atmosphere with 5% CO_2_ at 37 °C.

### 4.3. Western Blot Analysis

Protein extracts were obtained using a lysis buffer containing Triton 1% and 1X cOmplete^TM^ Mini protease inhibitor cocktail (Merck-Roche, Darmstadt, Alemania), as previously described [76]. Protein concentrations were determined using the Pierce^TM^ BCA Protein Assay reagent, following the manufacturer’s instructions (Thermo Fisher Scientific, Waltham, MA, USA). Proteins (80 µg per lane) were separated by SDS–PAGE in 10% mini-gels (Bio-Rad, Hercules, CA, USA) and transferred to nitrocellulose membranes, as previously described [77]. Blots were blocked with 5% skim milk in PBS-0.1% Tween 20 and then probed with different primary antibodies. Mouse monoclonal anti-Caspase 3 (sc-56053), anti-Caspase 9 (sc-133109), anti-PARP-1 (sc-136208), anti-p53 (sc-47698), and anti-HO-1 (sc-136960) antibodies were from Santa Cruz Biotechnology (Santa Cruz, CA, USA); mouse monoclonal anti-β-Actin (A5316) was from Merck-Sigma-Aldrich, and rabbit polyclonal anti-Survivin (AF886) was from R&D systems (Minneapolis, MN, USA). The bound primary antibodies were detected with horseradish peroxidase-conjugated donkey anti-mouse (SA1-100) or goat anti-rabbit (GTX213110-01) secondary antibodies purchased from Thermo Fisher Scientific and Genetex (Irvine, CA, USA), respectively. Following incubation with the EZ-ECL reagent to detect HRP activity (Biological Industries), images were obtained using an ImageQuant LAS500 imager (General Electric, Uppsala, Sweden).

### 4.4. Cytotoxicity Assays

#### 4.4.1. Cell Proliferation

Cell proliferation was determined using the non-radioactive CellTiter 96^®^ AQueous One Solution Cell Proliferation Assay (MTS) (Promega, Madison, WI, USA). Gastric cells were seeded at a density of 5 × 10^3^ cells/well in a 96-well plate and following 24 h of culture, the cells were treated with different concentrations of *L. rivularis* flower extracts (EtOAc, DCM, Hex, or EtOH) ranging from 2.5 µg/mL up to 50 µg/mL for 24 h or 48 h, as indicated. Cells were also treated with purified leptocarpin (2.5–25 µg/mL) or Etoposide 50 µM as controls. At the end of the experiment, cells were incubated with 100 uL of fresh medium, including the indicated volume of MTS solution at 37 °C for 1 h. Finally, absorbance was determined at 490 nm in a TECAN infinite M200 PRO reader.

#### 4.4.2. Trypan Blue Exclusion Assay

Gastric cells were seeded at a density of 5 × 10^4^ cells/well in 12-well plates. On the next day, cells were treated with *L. rivularis* flower extracts (DCM 5 µg/mL, EtOAc 5 µg/mL, Hex 10 µg/mL or EtOH 15 µg/mL), leptocarpin 5 µg/mL or Etoposide 50 µM as a control for 24 h or 48 h. Cell integrity was evaluated using the trypan blue exclusion assay [78]. The loss of membrane integrity can be measured using this vital dye, which cannot enter healthy living cells but is taken up by dying cells with permeabilized plasma membranes. Following treatment, cells were trypsinized, incubated with trypan blue 0.4% (Thermo Fisher Scientific), and counted in a Neubauer chamber.

#### 4.4.3. DEVDase Activity

Gastric cells were seeded at a density of 8 × 10^5^ cells/well in a 60 mm plate and cultured for 24 h. Then, the cells were treated with *L. rivularis* flower extracts (DCM 5 µg/mL, EtOAc 5 µg/mL, Hex 10 µg/mL, or EtOH 15 µg/mL) for an additional 24 h period. Cell lysates were obtained as previously described [79]. Caspase-3/7 activity (DVEDase activity) was quantified by following the release of the fluorescent dye 7-amino-4-trifluoromethylcoumarin (AFC) from the substrate Asp-Glu-Val-Asp-AFC (Enzo Life Sciences, Lörrach, Germany) in the presence of 5–25 µg of lysates in a final volume of 250 µL of reaction. Fluorescent emission (λexc = 375 nm, λem = 530 nm) was followed with an Infinite M200Pro (TECAN) multiplate reader. As previously described, a unit of enzymatic activity is defined as 1 mmol of substrate transformed per min per mg of protein extract [79].

#### 4.4.4. Determination of Mitochondrial Transmembrane Potential

Gastric cells were incubated with the *L. rivularis* flower extracts (2.5 µg/mL and 10 µg/mL) for 24 h. FCCP 1 µM (carbonyl cyanide 4-(trifluoromethoxy) phenylhydrazone, Sigma-Aldrich) was included as a positive control [80]. Following treatment, cells were labeled with 1 μM Rhodamine 123 (Sigma-Aldrich) for 1 h at 37 °C. Next, cells were harvested by trypsinization and washed three times with chilled PBS. Fluorescence was determined by flow cytometry using the FL1 channel (FITC). Data are expressed as the percentage of Rhodamine 123-positive cells.

#### 4.4.5. LIVE/DEAD^®^ Viability/Cytotoxicity Assay

Gastric cells were incubated at a density of 5 × 10^4^ cells/well in a 12-well plate and cultured for 24 h. Then, cells were treated with *L. rivularis* flower extracts (DCM 5 µg/mL, EtOAc 5 µg/mL, Hex 10 µg/mL or EtOH 15 µg/mL), leptocarpin 5 µg/mL or Etoposide 50 µM for 24 h. Cell viability was determined using the LIVE/DEAD^®^ Viability/cytotoxicity kit (Molecular Probes, Eugene, OR, USA), following the manufacturer’s instructions.

#### 4.4.6. Clonogenic Assay

The clonogenic capacity of gastric cells was evaluated as previously described [81]. Briefly, cells were cultured for 24 h in complete RPMI 1640 and then treated with the *L. rivularis* flower extracts (DCM 5 µg/mL, EtOAc 5 µg/mL, Hex 10 µg/mL, or EtOH 15 µg/mL), leptocarpin 5 µg/mL or Etoposide 50 µM for 24 h. Following treatment, cells were trypsinized, and cell viability was determined using the trypan blue exclusion assay. Trypan blue-negative cells were reseeded at a density of 500 cells per well in a 6-well plate and cultured for eleven days. The colonies were fixed with 100% methanol for 15 min at room temperature, stained with 0.5% crystal violet for 30 min, photographed, and counted.

### 4.5. Detection of Senescent Cells

Gastric cells were seeded on glass coverslips at a density of 5 × 10^4^ cells/well in a 24-well plate and cultured for 24 h. On the next day, the cells were treated with the *L. rivularis* extracts (DCM 5 µg/mL, EtOAc 5 µg/mL, Hex 10 µg/mL, or EtOH 15 µg/mL) or Etoposide 25 µM for 72 h. The apparition of senescent cells was evaluated using the CellEventTM Senescence Green Detection Kit (Invitrogen, Waltham, MA, USA), which is based on detecting intracellular accumulation of a fluorescent product (excitation/emission at 490/514 nm) as a consequence of β-galactosidase activity, a recognized biomarker that appears overexpressed in senescent cells [82,83,84]. Nuclei were counterstained with DAPI 1µg/mL (Thermo Fisher Scientific).

### 4.6. Cell Migration and Invasion Assays

For cell invasion assays, gastric cells were treated with the *L. rivularis* flower extracts (DCM 5 µg/mL, EtOAc 5 µg/mL, Hex 10 µg/mL, or EtOH 15 µg/mL), leptocarpin 5 µg/mL or Etoposide 50 µM for 24 h. Subsequently, a quantity of 5 × 10^4^ trypan blue-negative gastric cells was resuspended in 300 µL of serum-free RPMI 1640 medium and seeded in a Corning^®^ Transwell^®^ polycarbonate membrane cell culture insert (8.0 µm), coated with 20 µL of Matrigel^®^ Growth Factor Reduced Basement Membrane Matrix (Corning, Santa Barbara, CA, USA), as previously described [85]. Additionally, a volume of 0.5 mL of RPMI 1640 medium containing 3% fetal bovine serum was added to the lower chamber as a chemoattractant. For migration assays, gastric cells were treated similarly and seeded at a density of 3 × 10^4^ cells per insert, but without the addition of Matrigel. Following 24 h or 48 h of incubation for migration and invasion assays, respectively, the upper surface of the insert’s porous membrane was wiped off with a cotton swab. Cells that invaded the lower surface of the permeable membrane were fixed with 100% methanol for 15 min and stained with 0.5% crystal violet in 25% methanol/PBS solution for 30 min. The stained cells were counted in five random fields per filter (total magnification 400×), using a total of three filters (n = 3). Invasion/migration is shown as the percentage of treated cells found on the lower side of the insert membrane, compared with the number of untreated control cells (100%).

### 4.7. Matrigel Vasculogenic Assay

EA.hy926 cells were seeded at a density of 4 × 10^5^ cells in 60 mm plates and cultured in Iscove′s modified Dulbecco′s medium (ThermoFisher Scientific, Waltham MA, USA), supplemented with 10% fetal bovine serum for 24 h. Then, the cells were treated with the EtOAc extract (1.0 and 2.5 µg/mL) or leptocarpin (1.0 and 2.5 µg/mL) for 24 h. Following treatment, cells were washed three times with PBS and maintained in DMEM F-12 Ham medium without serum for 3 h. Subsequently, cells were harvested in DMEM Ham F12 medium without serum and seeded at a density of 3 × 10^3^ cells/well in 24-well plates coated with 150 µL/well of Matrigel (Corning 356231, Glendale AZ, USA). As a positive control of angiogenesis, untreated EA.hy926 cells were cultured similarly in the presence of VEGF 20 ng/mL. Finally, cells were photographed after 24 h, and the angiogenic index was calculated based on the quantification of several morphological parameters described by Garrido M. et al. [47].

### 4.8. Statistical Analysis

All data are expressed as the mean ± standard deviation (S.D.) of at least three independent experiments. Data were processed using INSTAT v. 3.05 (GraphPad Software, San Diego, CA, USA, www.graphpad.com). In all cases, a value *p* < 0.05 (or less) determined with the non-parametric Kruskal–Wallis test and Mann–Whitney post-test was considered statistically significant.

## 5. Conclusions

In conclusion, this study offers essential evidence highlighting the antitumor effect of *L. rivularis* flower extracts on gastric cancer cells. The flower extracts that were analyzed demonstrated cytotoxicity, apoptotic cell death induction, senescence induction, migration and invasion inhibition, and angiogenesis decrease, suggesting that *L. rivularis* flowers might constitute promising sources of antitumoral bioactive compounds for pharmaceutical industries.

## Figures and Tables

**Figure 1 ijms-24-01439-f001:**
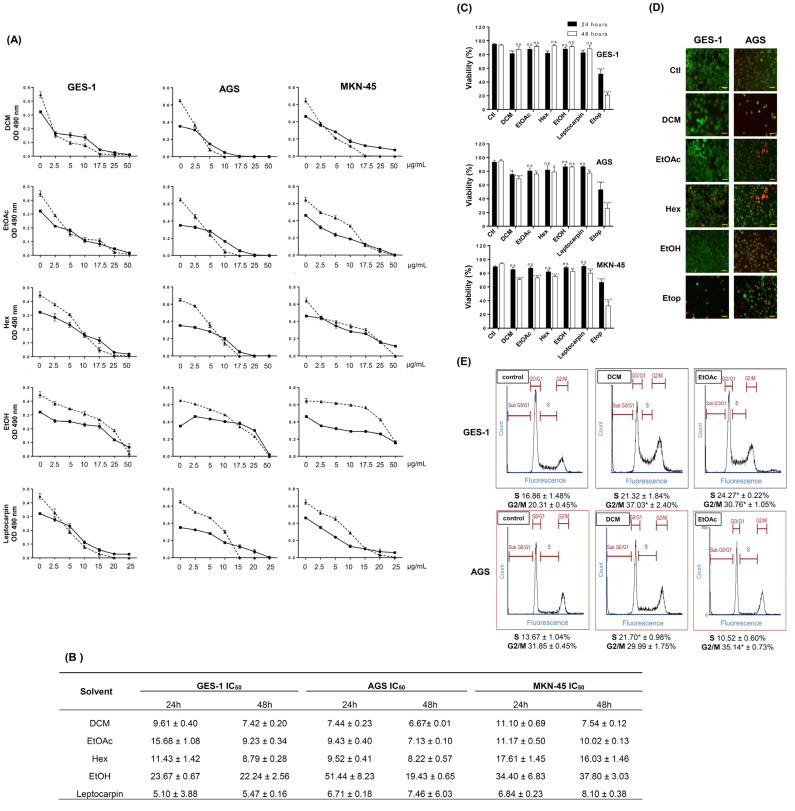
Antiproliferative effect of *L. rivularis* flower extracts on gastric cell lines. (**A**) Gastric cancer (AGS and MKN-45) and the normal immortalized (GES-1) cell lines were treated with different concentrations of extracts (0, 2.5, 5, 17.5, 25, and 50 µg/mL) or leptocarpin (0, 2.5, 5, 10, 15, 20, and 25 µg/mL) for 24 h (circles) or 48 h (triangles), respectively. Proliferation was evaluated with the MTS assay. Curves represent the absorbance readings at 490 nm from at least three independent experiments (means ± S.D.). (**B**) The table shows the half-maximal inhibitory concentration (IC_50_) calculated for each extract (DCM, EtOAc, Hex, and EtOH) or leptocarpin treatments in GES-1, AGS, and MKN-45 cells, using a fitted second-order polynomial curve (means ± S.D., n = 3). (**C**) GES-1, AGS, and MKN-45 cells were treated with DCM 5 µg/mL, EtOAc 5 µg/mL, Hex 10 µg/mL, EtOH 15 µg/mL, Etoposide (Etop 50 µM), or leptocarpin 5 µg/mL for 24 h (black bars) and 48 h (white bars). Cell viability was evaluated using the trypan blue exclusion assay. Bars represent the percentage of living cells identified as trypan blue-negatives, compared with untreated cells (means ± S.D., n = 3, * *p* ≤ 0.05, ** *p* ≤ 0.01, *** *p* ≤ 0.001, and **** *p* ≤ 0.0001, ns: non-significant). (**D**) GES-1 and AGS cells were treated with DCM 5 µg/mL, EtOAc 5 µg/mL, Hex 10 µg/mL, EtOH 15 µg/mL or Etoposide 50 µM for 24 h. Then, live/dead cells were labeled using the Calcein-AM (green fluorescence)/EthD-1 (red fluorescence) combination. A representative result is shown. Magnification bar= 100 µm. (**E**) GES-1 and AGS cells were treated with the extracts (DCM 5 µg/mL and EtOAc 5 µg/mL) or 0.1% ethanol as a control for 24 h. Cell cycle distribution was determined by flow cytometry. Representative profiles of the cell cycle distribution for the indicated conditions are shown. The percentage of cells in the S and G2/M phases is indicated at the bottom of each image (means ± S.D., n = 3; * *p* < 0.05).

**Figure 2 ijms-24-01439-f002:**
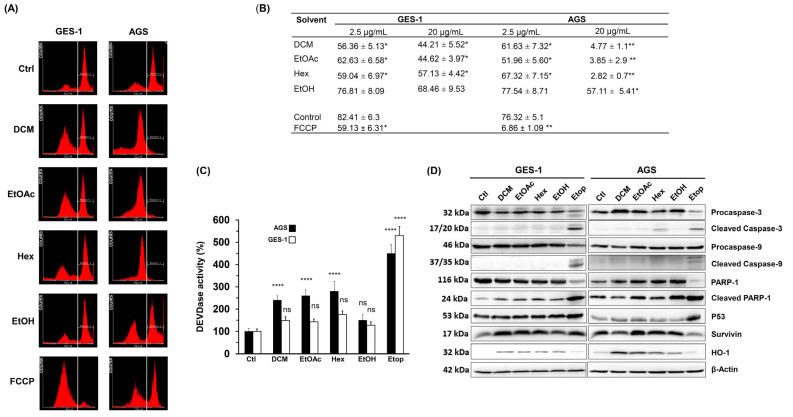
*L. rivularis* flower extracts trigger loss of mitochondrial membrane potential and apoptosis in gastric cancer cells. (**A**) GES-1 and AGS cells were treated with the extracts (DCM, EtOAc, Hex, or EtOH) at 20 µg/mL or FCCP 1 µM for 24 h. The cells were then stained with Rhodamine 123 and analyzed by flow cytometry. Representative histograms are shown. The vertical white line separates Rhodamine-positive (right side) and Rhodamine-negative (left side) cell populations. (**B**) GES-1 and AGS were treated with *L. rivularis* flower extracts at concentrations of 2.5 µg/mL and 20 µg/mL. The percentage of Rhodamine-positive cells is shown (means ± S.D., n = 3; * *p* < 0.05, and ** *p* < 0.01). (**C**,**D**) GES-1 and AGS cells were treated with DCM 5 µg/mL, EtOAc 5 µg/mL, Hex 10 µg/mL, EtOH 15 µg/mL, or Etoposide 50 µM for 24 h. (**C**) DEVDase activity was evaluated in total cell lysates. Bars represent the percentage of DEVDase activity compared to the control condition (means ± S.D., n = 3, **** *p* ≤ 0.0001, ns: non-significant). (**D**) Changes in the protein levels of Procaspase-3/cleaved Caspase-3, Procaspase-9/cleaved Caspase-9, PARP-1/cleaved PARP-1, p53, Survivin, and HO-1 were evaluated by Western blot analysis of protein extracts. Representative blots are shown.

**Figure 3 ijms-24-01439-f003:**
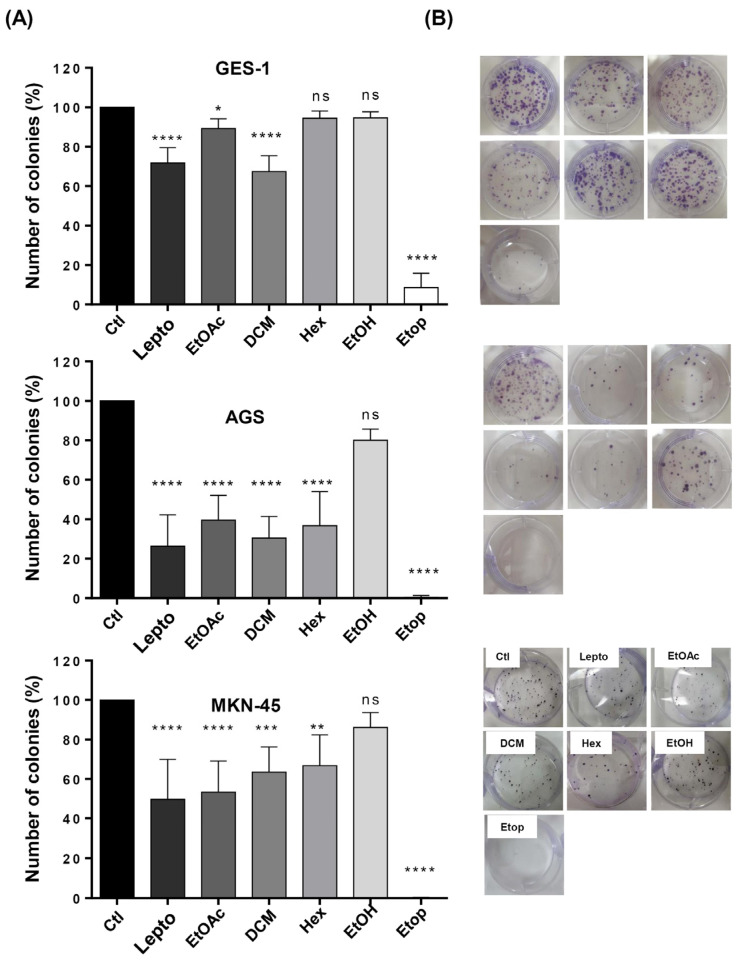
Effect of *L. rivularis* flower extracts on the clonogenic potential of gastric cells. Synchronized GES-1, AGS, and MKN-45 cells were treated with *L. rivularis* flower extracts (DCM 5 µg/mL, EtOAc 5 µg/mL, Hex 10 µg/mL, or EtOH 15 µg/mL), Etop 50 µM (Etop) or leptocarpin (Lepto) 5 µg/mL for 24 h. Then, the clonogenic potential of gastric cells was evaluated as described in the Material and Methods section. (**A**) Bars represent the percentage of colonies formed in response to the different treatments, compared with the control condition (means ± S.D., n = 3, * *p* < 0.05, ** *p* < 0.01, *** *p* ≤ 0.001, and **** *p* ≤ 0.0001, ns: non-significant). (**B**) Representative images of the clonogenic assay in GES-1, AGS, and MKN-45 cells treated with the extracts, leptocarpin, or Etoposide in standard 6-well plates.

**Figure 4 ijms-24-01439-f004:**
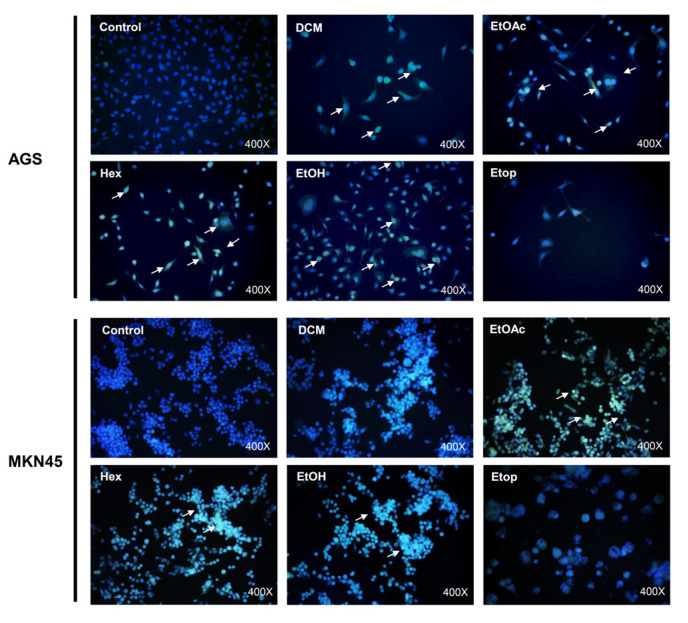
*L. rivularis* flower extracts induce senescence in gastric cancer cells. AGS and MKN-45 cells were treated with *L. rivularis* flower extracts (DCM 5 µg/mL, EtOAc 5 µg/mL, Hex 10 µg/mL, or EtOH 15 µg/mL) or Etoposide 25 µM for 72 h. The cells were then incubated with CellEvent Senescence Green β-gal substrate at pH 5 for 2 h at 37 °C, washed with PBS, and mounted with Fluoromount + DAPI 0.5 µM. White arrows indicate cells with a positive green signal in representative images.

**Figure 5 ijms-24-01439-f005:**
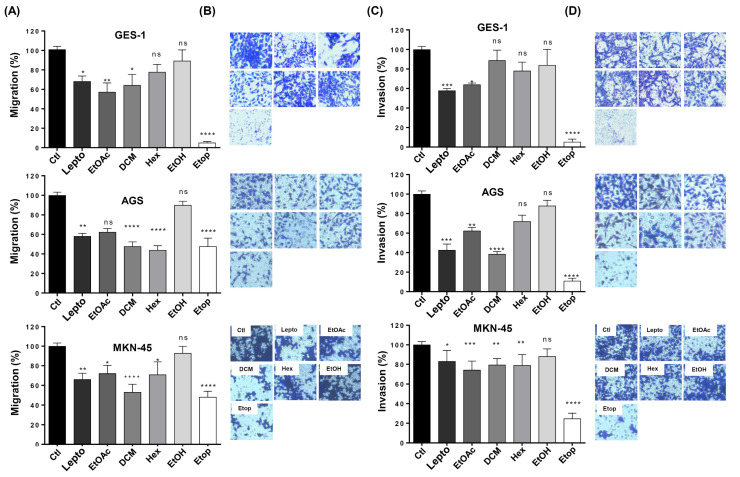
Effect of *L. rivularis* flower extracts on the migration and invasion potential of gastric cells. GES-1, AGS, and MKN-45 gastric cells were treated with the extracts (DCM 5 µg/mL, EtOAc 5 µg/mL, Hex 10 µg/mL, or EtOH 15 µg/mL), leptocarpin (Lepto) 5 µg/mL, or Etoposide 50 µM (Etop) for 24 h and then harvested and seeded into the upper compartment of the Boyden’s migration/invasion chamber (see Material and Methods section). The number of cells that moved into the lower chamber represented the migrating/invading cells. (**A**,**C**) Bars represent the percentage of the migration (**A**) or invasion (**C**) of GES-1, AGS, and MKN-45 cells treated with the indicated extracts, compared to the control cells (100% migration/invasion). Data represent means ± S.D. of three independent experiments (* *p* < 0.05, ** *p* < 0.01, *** *p* ≤ 0.001, and **** *p* ≤ 0.0001, ns: non-significant). (**B**,**D**) Representative pictures of the migration (**B**) and invasion (**D**) assays in GES-1, AGS, and MKN-45 cells are shown (magnification × 100).

**Figure 6 ijms-24-01439-f006:**
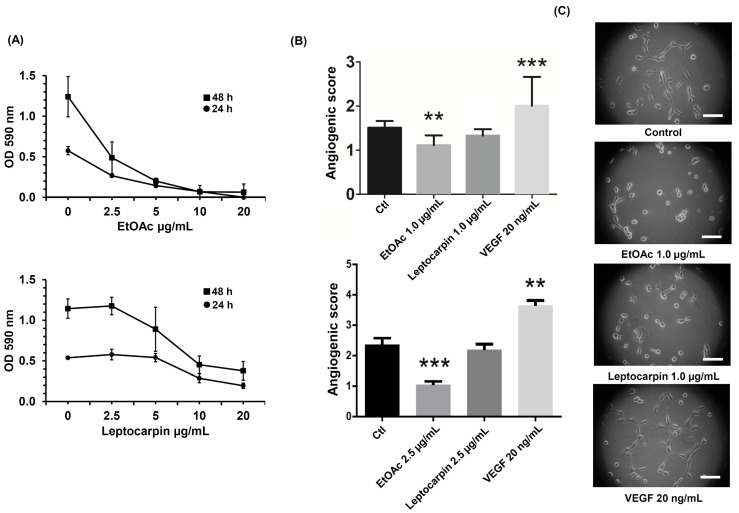
The EtOAc *L. rivularis* flower extract and leptocarpin reduce the vasculogenic capacity of EA.hy926 cells. (**A**) EA.hy926 cells were treated with different concentrations of the EtOAc extract or leptocarpin (0, 2.5, 5, 10, and 20 µg/mL) for 24 h (circles) or 48 h (squares). Cell proliferation was evaluated with the MTS assay. Dose–response curves represent absorbance values at 490 nm from at least three independent experiments (means ± S.D.). (**B**) EA.hy926 cells were treated with the EtOAc extract (1.0 µg/mL and 2.5 µg/mL) or leptocarpin (1.0 µg/mL and 2.5 µg/mL) for 24 h and then seeded in Matrigel-coated plates and cultured for an additional period of 24 h. Bars represent the angiogenic scores obtained for each condition. Values represent the means ± S.D. from three independent experiments (** *p* < 0.01, and *** *p* ≤ 0.001 vs. the control condition). (**C**) Representative images of EA.hy926 cells cultured for 24 h on Matrigel-coated plates following EtOAc 1.0 µg/mL, leptocarpin 1.0 µg/mL, or VEGF 20 ng/mL treatments. Magnification bar = 50 μm.

**Table 1 ijms-24-01439-t001:** Majority composition of Hex extract of *L. rivularis* flowers.

N° Peak	RT (min)	Main Components	RI ^a^	RI Ref ^b^	%Area ^c^	Match	Identification
1	22.81	Grosheiminok	2143	2143	1.58	850	RI, MS, Co
2	23.28	(*Z*)-18-Octadec-9-enolide	2155	2158	0.52	950	RI, MS
3	26.46	Reynosin	2266	2266	1.35	900	RI, MS, Co
4	31.66	duvatrienediol	2397	2400	25.70	860	RI, MS
5	34.62	3-Ethyl-5-(2-ethylbutyl)octadecane	2410	2413	3.89	930	RI, MS
6	34.81	Tetraneurin D	2490	2494	0.69	970	RI, MS
7	35.62	2-Methylenecholestan-3-ol	2650	2652	0.41	970	RI, MS
8	35.86	*α*-Tocopherol	3149	3149	0.35	950	RI, MS, Co
9	37.69	Stigmasterol	3170	3170	0.70	940	RI, MS, Co
10	40.50	γ-Sitosterol	3290	3290	0.33	930	RI, MS, Co
11	42.69	ß-Amyrin	3330	3337	0.35	850	RI, MS
12	43.79	ß-Amyrin acetate	3333	3339	4.78	910	RI, MS
13	44.57	Lupeyl acetate	3380	3380	10.96	850	RI, MS, Co
14	46.06	Betulinaldehyde	3630	3629	0.13	820	RI, MS
15	46.62	1-Heptatriacontanol	3941	3942	0.46	900	RI, MS

^a^ RI: retention indices relative to C_8_–C_36_ *n*-alkanes on the HP-5 MS capillary column; ^b^ RI: retention index from the literature [33]; ^c^ surface area of GC peak; MS: comparison of the mass spectra with those of the NIST 14; Co: co-elution with standard compounds available in our laboratory.

**Table 2 ijms-24-01439-t002:** Majority composition of DCM extract of *L. rivularis* flowers.

N° Peak	RT (min)	Main Components	RI ^a^	RI Ref ^b^	%Area ^c^	Match	Identification
1	12.13	Intermedeol	1660	1675	2.94	810	RI, MS
2	16.03	Dehydrocostus lactone	1860	1866	2.76	890	RI, MS, Co
3	25.66	(Z)-18-Octadec-9-enolide	2190	2208	2.57	980	RI, MS
4	26.48	Reynosin	2260	2266	1.05	860	RI, MS, Co
5	29.90	Tricosane	2330	2300	3.85	970	RI, MS
6	33.95	Thunbergol	2500	2498	8.82	940	RI, MS
7	23.18	2-Methylenecholestan-3-ol	2650	2652	1.15	920	RI, MS
8	36.05	Triacontane	3000	3000	4.36	870	RI, MS
9	36.34	ß-Amyrin	3315	3337	0.43	940	RI, MS
10	36.96	α-Amyrin	3360	3376	0.98	940	RI, MS
11	37.59	Tetratriacontane	3400	3400	6.39	860	RI, MS, Co
12	44.48	ß -amyrin acetate	3430	3437	6.43	910	RI, MS
13	44.59	Lupeol	3494	3486	0.25	930	RI, MS, Co
14	45.53	lupeyl acetate	3530	3525	14.96	900	RI, MS
15	45.67	Betulinaldehyde	3743	3760	0.70	900	RI, MS
16	46.62	1-Heptatriacotanol	3940	3942	6.71	880	RI, MS, Co
17	50.52	Oleic acid 3-(octadecyloxy)propyl ester	4100	4149	0.76	850	RI, MS, Co

^a^ RI: retention indices relative to C_8_–C_36_ *n*-alkanes on the HP-5 MS capillary column; ^b^ RI: retention index from the literature [33]; ^c^ surface area of GC peak; MS: comparison of the mass spectra with those of the NIST 14; Co: co-elution with standard compounds available in our laboratory.

**Table 3 ijms-24-01439-t003:** Majority composition of EtOAc extract of *L. rivularis* flowers.

No.	RT (min)	Main Components	RI ^a^	RI Ref ^b^	% Area ^c^	Match	Identification
1	9.575	Thujone	1095	1089	6.46	927	RI, MS, Co
2	12.750	Isopentenyl acetate	1196	1200	2.87	950	RI, MS
3	13.215	*trans*-Sabinyl acetate	1265	1264	4.01	934	RI, MS
4	18.110	Intermedeol	1660	1667	3.44	949	RI, MS
5	18.234	6-*epi*-Shyobunol	1883	1881	3.59	953	RI, MS
6	18.450	Eicosane	2000	2000	2.87	989	RI, MS
7	18.610	Geranyllinalool	2040	2046	2.08	910	RI, MS
8	19.85	Leptocarpin	2050	2050	3.46	967	RI, Co
9	20.120	Duvatrienediol	2395	2400	6.20	960	RI, MS
10	21.250	Linalool	2486	2491	2.26	899	RI, MS, Co
11	21.590	2-Methylenecholestan-3-ol	2651	2652	2.32	956	RI, MS
12	42.06	Stigmasterol	3140	3142	24.02	940	RI, MS, Co
13	42.70	Fenretinide	3290	3289	13.76	930	RI, MS, Co
14	43.86	1-Heptatriacontanol	3938	3942	8.49	900	RI, MS

^a^ RI: retention indices relative to C_8_–C_36_ *n*-alkanes on the HP-5 MS capillary column; ^b^ RI: retention index from the literature [33]; ^c^ surface area of GC peak; MS: comparison of the mass spectra with those of the NIST 14; Co: co-elution with standard compounds available in our laboratory.

**Table 4 ijms-24-01439-t004:** Selectivity index (SI) for *L. rivularis* flower extracts in AGS and MKN45 gastric cancer cells. The selectivity index is the ratio of the IC_50_ values of the treatments on GES-1 cells to those in the gastric cancer cell lines.

Solvent	AGS SI		MKN45 SI	
	24 h	48 h	24 h	48 h
DCM	1.29	1.11	0.86	0.98
EtOAc	1.66	1.29	1.40	0.92
Hex	1.20	1.07	0.64	0.54
EtOH	0.46	1.15	0.68	0.58
Leptocarpin	0.76	0.68	0.74	0.62

## Data Availability

Not applicable.

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
