# Peer review of "Antitumoral Activity of *Leptocarpha rivularis* Flower Extracts against Gastric Cancer Cells"

_ijms, 2023, doi:10.3390/ijms24021439_

Round 1
Reviewer 1 Report
Maybe a representative MS spectrum of an extract.
Concentration effects did not seem to be discussed much.
From line 140 to 155 is more of a introduction or discussion discourse. Maybe move to the appropriate section
It was said that the extracts were rotovapted to dryness. How were the extracts reconstituted after storage. Did not see that in the methods section.
Control preparation in the method section?
Author Response
Replies for Reviewer 1
We thank Reviewer 1 for the positive appreciation of our manuscript
Comments and Suggestions for Authors
- Maybe a representative MS spectrum of an extract.
Response: We think it is unnecessary to place the MS because each table has the match quality specified. Our analysis of unknown substances in L. rivularis extracts by GC-MS is based on identifying compounds by spectral comparisons with mass spectral libraries. Thus, we use the NIST database, where the EI mass spectra fragmentation pattern is compared by matching spectra with reference mass spectra from the library. If the match factors from both sources give very similar results (scores above 900) are considered excellent candidates, while all compounds that present scores between 800-900 have "good" to "excellent" spectral matches with the NIST library spectra.
Another analytical control for GC-MS is the Kovats index. We use retention indices relative to the C8-C36 n-alkanes on the HP-5 MS capillary column, as indicated for each table. In addition, we compare with an RI retention index from the literature. Kovats (retention) indices express the number of carbon atoms (multiplied by 100) of a hypothetical n-alkane that would have a retention volume (time) identical to that of the peak of interest when analyzed under the same conditions. Finally, we used standards of known molecules that were co-injected to corroborate their presence in the extracts. With these parameters, we performed each analysis indicated in each table. We appreciate your comments and indications.
- Concentration effects did not seem to be discussed much.
Response: We have included a brief discussion in this respect.
- From lines 140 to 155 is more of an introduction or discussion discourse. Maybe move to the appropriate section
Response: You are right. We moved these sentences to the discussion section.
- It was said that the extracts were rotovapted to dryness. How were the extracts reconstituted after storage? Did not see that in the methods section.
Response: Thanks; this detail was not noticed; now, we include the information in the methodology section.
- Control preparation in the method section?
Response: Thanks; this detail was not noticed; now, we include the information in the methodology section.
Best regards
MAVV

Reviewer 2 Report
Manuscript provided by Carrasco et al. (ijms-2095089) includes studies of anti -cancer activity of floral extracts of Leptocarpha rivularis in relation to gastric cancer cells in vitro. In their research, the authors use four extracts prepared on the basis of various extractants. Two gastric cancer cell lines are used as a research model: AGS and MKN-45, as well as the human gastric epithelial GES-1 cell line as a control. The tests include an analysis of cytotoxicity of extracts on the tested cell lines, assessment of spread potential, proliferation and senescence. The reviewer did not find any other studies that would investigate the effect of L. rivularis extracts on gastric cancer cell lines, so the topic undertaken by the authors is innovative and new. References are well matched to the topic presented. The work would be an interesting source of new information for IJMS readers. However, the reviewer has a few remarks and comments that require clarification and possible specification.
Major:
1. 1. The reviewer has great doubts about the rather high toxicity of the presented extracts on gastric cancer cell line in relation to the normal GES-1 line. The reviewer suggests presenting the selectivity index value (SI) for the tested extracts on individual cell lines. In previous studies carried out by the authors of the manuscript in which they used the same L. rivularis extracts, the SI was calculated in relation to the cancer cell lines tested there (https://doi.org/10.3390/molecules26010067).
2. 2. The reviewer cannot read the data from the graphs used in Figures: 1A, 1E, 2B. The reviewer recommends improving the quality of figures or enlarging them.
3. The reviewer has reservations about the interpretation of the cell cycle described in the results section, but is unable to decipher which plot corresponds to which sample in Figure 1E. In addition, in the description of the Figure 1E it is stated that each line was treated with 4 extracts and a control sample was present. And in Figure 1E there are only 3 histograms for each cell line.
Minor:
1. In figures 3A, 5A, 5C there are no captions on the x-axis of the graphs.
Kind regards.
Author Response
Comments and Suggestions for Authors
Manuscript provided by Carrasco et al. (ijms-2095089) includes studies of anti -cancer activity of floral extracts of Leptocarpha rivularis in relation to gastric cancer cells in vitro. In their research, the authors use four extracts prepared on the basis of various extractants. Two gastric cancer cell lines are used as a research model: AGS and MKN-45, as well as the human gastric epithelial GES-1 cell line as a control. The tests include an analysis of cytotoxicity of extracts on the tested cell lines, assessment of spread potential, proliferation and senescence. The reviewer did not find any other studies that would investigate the effect of L. rivularis extracts on gastric cancer cell lines, so the topic undertaken by the authors is innovative and new. References are well matched to the topic presented. The work would be an interesting source of new information for IJMS readers. However, the reviewer has a few remarks and comments that require clarification and possible specification.
Response: We thank Reviewer for the positive appreciation of our manuscript.
Major:
- The reviewer has great doubts about the rather high toxicity of the presented extracts on gastric cancer cell line in relation to the normal GES-1 line. The reviewer suggests presenting the selectivity index value (SI) for the tested extracts on individual cell lines. In previous studies carried out by the authors of the manuscript in which they used the same L. rivularis extracts, the SI was calculated in relation to the cancer cell lines tested there (https://doi.org/10.3390/molecules26010067).
Response: As requested, we have included a table including SI values for MKN45 and AGS cells (table 3). Important to mention that the extracts affected proliferation and spreading parameters in all gastric cell lines tested; however, these induced the loss of mitochondrial membrane potential and apoptosis selectively in gastric cancer cells. The selectivity index is calculated based on IC50 values from MTS proliferation assays, which do not discriminate between antiproliferative or lethal events. Therefore, our conclusions were not based on SI values.
- The reviewer cannot read the data from the graphs used in Figures: 1A, 1E, 2B. The reviewer recommends improving the quality of figures or enlarging them.
Response: This was amended as requested.
- The reviewer has reservations about the interpretation of the cell cycle described in the results section, but is unable to decipher which plot corresponds to which sample in Figure 1E. In addition, in the description of the Figure 1E it is stated that each line was treated with 4 extracts and a control sample was present. And in Figure 1E there are only 3 histograms for each cell line.
Response: The figures have been improved, and the legend has been corrected. We use all the extracts to analyze the changes in the cell cycle distribution. However, we only show a couple of extracts to simplify the figure.
Minor:
- In figures 3A, 5A, 5C there are no captions on the x-axis of the graphs.
Response: This has been corrected as requested.
Kind regards.

Round 2
Reviewer 2 Report
The authors responded to all the observations made during the reviewing process. I recommend the manuscript to be published.